# Toward Tourists–Media–Cities Tourism: Xi'an as a Wanghong City

Tingting Fan and Zhongxuan Lin *

School of Journalism and Communication, Jinan University, Guangzhou 510632, China
* Correspondence: zxlin@jnu.edu.cn

**Abstract:** This study investigated the phenomenon of wanghong cities in China to illustrate the dynamic relationships between media, tourists, and cities in the new normal of the post-COVID-19 era. Specifically, this study proposes the innovative analytical framework of tourists–media–cities ("ToMeCi"), which is grounded in tourism studies, media studies, and urban studies, but goes beyond the traditional concepts and previous studies of media and city, tourism and media, and tourism and city. Based on a case study of Xi'an, one of the most famous wanghong cities in China, this study attempted to answer the following research questions: how can the deployment of creative media practices create new digital tourism patterns in the specific Chinese context of wanghong cities; moreover, why is this reasonable and possible, and what are the implications? For the purpose of this study, we employed a qualitative research method and conducted online and offline ethnographic fieldwork, textual analysis, and in-depth interviews with 22 tourists and 26 short-video producers or live streamers. The findings reveal that the city of Xi'an was transformed into a wanghong city to attract tourists, who interact with the city through specific media practices of clocking in and live streaming, with a new digital tourism model of the cyberflaneur emerging against the specific backdrop of the COVID-19 pandemic. Finally, we discuss the possible contributions and limitations of the phenomenon.

**Keywords:** COVID-19; cyberflaneur; mediatization; tourism; ToMeCi; wanghong city

## 1. Introduction

In January 2018, the business district of Xi'an, called Yongxing Square, launched an entertainment program called "bowl-smashing wine" (摔碗酒), in which tourists spent five yuan to drink a bowl of wine and then smash the bowl on the ground. This program made the place a spot for "clocking in" (da ka, 打卡). "Clocking in" means signing in at work, but now it has become a way for tourists physically visiting certain places to mark events, persons, or things using media, especially via smartphones. In the case of "bowl-smashing wine" (摔碗酒), thousands of tourists went to Xi'an to experience this program, shoot videos, and upload them to the Internet. This made "bowl-smashing wine" (摔碗酒) a popular hashtag on TikTok, with 8.91 million short videos and 150 million viewers. Such creative media practices open up a new vision for urban tourism, as tourists' creativity with clocking in has also become an important part of tourism planning and transformation for cities. This kind of clocking in has become an important part of people's trips and a new pattern of tourism, which is especially crucial for wanghong cities, that is, cities that are famous online and thus become preferred tourist destinations offline, such as Xi'an, because tourists in these cities are usually influenced by other people's clocking in photos or videos [1].

Underpinned by diverse creative media practices, the emergence of wanghong cities in China began in 2018 [1]. Cities such as Xi'an, Chengdu, Chongqing, and Changsha have been labeled wanghong cities because they met the criteria by becoming famous online, thus tourists tend to choose them as travel destinations offline [2]. During this process, creative media practices such as clocking in have had a significant impact on the phenomenon of wanghong cities. It has already become common for tourists to share

their personal travel experiences on Weibo, Honeycomb, Fliggy, Facebook, and other social media platforms, and interact with followers and others who have the same travel interests. In this way, they can modify the image of a city through their personal media practices, creating and participating in urban tourism marketing activities [3]. Indeed, there are many studies on the relationship between media practices and urban tourism [4–7], especially on how popular social media platforms impact the cognition of the urban tourism image [8–10], manifested in the way tourists create images of destinations [11]. Govers, Go, and Kumar note that "technological advancement, the global media, and increased international competition affect the way in which destinations are imagined, perceived, and consumed. Image formation is no longer a one-way 'push' process of mass communication, but a dynamic one of selecting, reflecting, sharing, and experiencing" [11]. Recently, new media technologies such as social media, information and communications technology (ICT), augmented reality (AR), and virtual reality (VR) have become increasingly important for the tourism industry, not only for destination marketing but also in helping tourists to choose a destination and avoid risks [12–14].

Against the new backdrop of the COVID-19 pandemic, which has affected every part of the world, tourism, an environmentally sensitive industry, has been hit hard [15,16]. Researchers generally believe that the pandemic has hurt tourism because of the resultant lockdowns, isolation, quarantine, and travel restrictions [17,18]. Furthermore, nonpharmacological measures, such as limiting the flow of tourists, taking people's temperature, requiring masks, and displaying health codes in scenic spots, also affect tourist activities, especially in offline travel [19,20]. In the face of such challenges, new media practices based on technologies such as AR, VR, live streaming, and short videos have created new opportunities for the tourism industry and promoted new patterns, such as virtual tourism [21–23]. According to the data center of the Xi'an Municipal Administration of Culture and Tourism, during China's National Day holiday in 2021, from October 1 to 7, the number of domestic tourism trips had decreased by 1.5% from the previous year, and domestic tourism income decreased by 4.7%; however, many scenic spots in wanghong cities, such as Datang Everbright City in Xi'an, attracted more people than in previous years [24]. Therefore, the study of wanghong cities and the theoretical frameworks and practical mechanisms behind them may have some reference value for thinking about reopening scenic spots and restoring tourism as a whole in the period after COVID-19 [18].

Accordingly, this study is situated in a specific Chinese context, to examine the particular phenomenon of wanghong cities against the backdrop of the pandemic, and digital city tourism in the new normal in the post-COVID-19 era. In particular, this study takes Xi'an, one of the most famous wanghong cities in China, as a special case to illustrate the dynamic relationships between media, tourists, and cities. In doing so, an innovative analytical framework of tourists–media–cities ("ToMeCi") is proposed, which has its roots and paths in tourism, media, and urban studies, but which moves beyond the previous theoretical framework of the media city [25,26]. Based on the research methods of online and offline ethnographic fieldwork and in-depth interviews, this study examines the creative media practices of both the city (Xi'an) and tourists to illustrate how ToMeCi tourism works and why it is reasonable and possible, and what its implications are. In particular, the study demonstrates how the city interacts with tourists through media transformation, how tourists' media practices transform the city and the tourism industry, and how the wanghong city phenomenon has brought new possibilities for the whole tourism industry both during the pandemic and in the post-COVID-19 era. Finally, we discuss the possible contributions and limitations of the phenomenon.

## 2. Literature Review: Toward an Innovative Framework of ToMeCi Tourism

Focused on the new phenomenon of wanghong cities, this study developed a triadic theoretical framework of tourists–media–cities (ToMeCi) to rejuvenate the relationship between the three. In recent years, the important phenomenon of wanghong cities has attracted the interest of scholars in the fields of tourism studies, urban studies, and media

studies [27–29]. In this study, we aim to go beyond these traditional studies of the relationships between tourists and cities, media and cities, and tourists and media to further explore the dynamic interactivity within the innovative framework of ToMeCi, which has its roots and paths in urban tourism studies, as well as media and tourism studies. We summarize these previous studies below.

Studies on urban tourism focus on the relationship between tourism and cities, particularly the impact of tourism on cities [30,31]. On the one hand, tourism can promote urban economic development and infrastructure construction and improve the employment rate, which are positive effects [32]. On the other hand, having many tourists can result in urban traffic congestion, pollution, and other problems [29]. Studies of urban tourism also focus on tourists and locals, examining locals' views on tourism development and how visitors move around the city [33–35]. Overall, creative and historical cities such as Xi'an have a higher probability of becoming tourist destinations [36–38]. This kind of city tourism is growing rapidly in the tourism market because of rapid urbanization, low travel costs, an increase in short vacations, and the popularity of online booking [39].

Digital media is playing an increasingly important role in the development of urban tourism. Accordingly, studies of digital tourism have focused on the relationship between media, especially digital media, and tourism [40,41]. This strand can be traced back to traditional studies on how tourism is presented and promoted through texts, images, and videos [42–44], as well as more recent studies of how media technologies have become key factors in the competition among destinations. For example, location technology can help tourists navigate cities [45], VR technology can enhance the visitor experience [46], and ICT makes travel in cities more convenient [47]. Moreover, digital media and digital tourism are playing increasingly important roles in tourism studies against the new backdrop of the COVID-19 pandemic [48,49].

In addition to the relationship between tourism and cities or media, the wanghong cities phenomenon has also led to a relationship between media and cities, which is relatively scarcely studied in the fields of tourism and even urban tourism. However, the relationship between media and cities has long been a critical topic in media studies, with the media city being a specific research strand [26,50]. This strand of research assumes that the city is originally a medium [51] and that the contemporary city is a media–architecture complex [26]. From this perspective, scholars focus on topics such as urban digital infrastructure [52], mobile media in urban tourism [53], the impact of media on citizens [50,54,55], and geomedia in cities [50]. More recently, studies of the media city have continued to develop with contributions from more diverse disciplines, and can be roughly divided into two approaches: cities in media and media in cities.

The cities in media approach focuses mainly on media representations of cities; as Wearing explains, "Media representations of destinations play an important role in the attractiveness of tourism" [56]. Many scholars in different disciplines have discussed the important role of literary works, films, and other media in constructing images of cities and promoting tourist destinations; this highlights the media's representational power for cities [4]. Therefore, cities are often eager to create videos to show their charm [43], or are otherwise reliant on other traditional media to portray their attractiveness [57]. According to this approach, cities are texts that can be presented by the media; accordingly, a city in the media is a simulation or representation of the real city. For example, tourism catalogs, travel magazines, and websites are simulations of urban reality and even of already simulated environments [58,59]. However, other scholars have criticized this approach for being one-sided and ignoring participation, feedback, and communication from and among other stakeholders, including city residents, local businesses, and others [7]. In the era of new media, it is believed that social media plays a particularly important role in urban marketing by making cities popular destinations, because promoting destinations on social media is more interactive and thus more effective than distributing traditional printed travel brochures [60–62].

In comparison, the media in cities approach emphasizes the diverse roles and values of media outlets such as newspapers, advertising posters, video screens, mobile devices, and other new forms in cities [63,64]. There have long been studies that discuss the role of newspapers in organizing urban life, providing public services to citizens, reflecting the objective needs of citizens, helping to solve specific problems, and maintaining traditions and basic values [63,65]. For example, Matthews discussed the value of city newspapers as a regional media outlet in the aftermath of disasters [66]. More recently, with the development of ICT, some scholars have tried to go beyond the question of how ICT, not as a single medium, but as a whole, can change urban life and improve the quality of life for citizens in many areas [67,68]. They argue that the comprehensive use of advanced ICT can help solve problems related to the benefits of information, transportation, mobility, energy distribution, and security, and can help with urban development, decision-making, and promoting innovative economic development [69]. Intelligent digital technology has penetrated every corner of the city, integrating into it as a whole and restructuring its communication links [70,71].

Notably, in the context of urban tourism, media and ICT enable tourism destinations to market in a cost effective manner and provide more accurate services to tourists [72]; developing countries can also participate in global tourism competition [40]. New media devices such as smartphones not only have a significant impact on people's travel plans, choice of travel time, travel method, and destination [73], but also change their behaviors and moods [74]. For example, people can use smartphones and the Internet to obtain information about destinations and book hotels and attractions, which is a great advantage for physical tourism [40]. Therefore, tourists are increasingly reliant on mobile devices such as smartphones when traveling, because of their ubiquity, immediacy, personalization, and access to information [75]. Besides smartphones, people can also use VR/AR and other technologies for online travel [41], especially in the context of a global pandemic that seriously hinders traditional physical tourism [49].

However, previous studies have focused mainly on the binary relationships between media and cities, as well as media and city tourism, and ignored the increasingly important role of tourists. Tourists, however, remain an important component of tourism research, because they decide which attractions are worth seeing and sharing with other tourists. Especially in the era of social media, images of cities and the tourism industry are no longer constructed only by mass media, but rather depend on every social media user. This is especially true in the context of wanghong cities, which rely heavily on the social media clock-in culture, in which tourists endorse locations with their media practices [2]. On the one hand, people are keen to travel to wanghong cities because they are influenced by photos or videos of others clocking in on social media; on the other hand, clocking in at a city's popular Internet spots becomes an important part of a trip [1]. With tourists flocking to social media, certain cities will stand out among the others. They have become wanghong destinations for more and more people, whereby we can observe the increasing interactivity of tourists, media, and the city.

Therefore, focused on the new phenomenon of wanghong cities, this study goes beyond the traditional methods and frameworks of past studies to develop the triadic theoretical framework of ToMeCi, rejuvenating the relationship between tourists, media, and cities. Indeed, in recent years, the important phenomenon of wanghong cities has attracted the interest of scholars in the fields of tourism studies, urban studies, and media studies [27,28,39]. Based on previous studies, this study further explores the dynamic interactivity of tourists–media–city, focusing on the following research questions: how does a city transform into a wanghong city to attract tourists? How do tourists interact with cities using specific media practices (e.g., clocking in and live streaming)? How have tourists traveled virtually (e.g., live streaming) in wanghong cities during the COVID-19 pandemic? What new digital tourism patterns (e.g., cyberflaneur) have emerged from these media practices, particularly against the specific backdrop of the COVID-19 pandemic?

## 3. Methods

Xi'an was observed over a long period of time, since the concept of the wanghong city was formed. We analyzed the city from the traditional theoretical perspective as a media city and developed a new analytical framework, ToMeCi. This study investigated the changes and innovations in Xi'an's tourism patterns during COVID-19 during two research periods. The first period was between December 2017 and May 2019; this was the key period when Xi'an transformed into a wanghong city. Our research in this phase used the methods of online and offline ethnographic fieldwork and in-depth interviews. We used purposive sampling instead of random sampling to choose our research field sites and interviewees, since the assumptions of qualitative research are different from those of quantitative research, as it is not aimed at generalizing social facts but at gaining an in-depth interpretation of cultural reality in a specific context [76]. Purposive sampling refers to "strategies in which the researcher exercises his or her judgment about who will provide the best perspective on the phenomenon of interest, and then intentionally invites those specific perspectives into the study" [76]. That said, this sampling strategy relies on the researchers' judgment about the people and sites that are critical to the phenomenon of interest in order to choose where and what to observe and whom to interview [77]. This rationale was further explained by Thomas Schwandt as follows: "Sites or cases are chosen because there may be a good reason to believe that 'what goes on there' is critical to understanding some process or concepts, or for testing or elaborating some established theory" [78].

Accordingly, the field sites we purposively chose were three of Xi'an's major landmarks: the city wall (城墙), Datang Everbright City (大唐不夜城), and Huimin Street (回民街). We conducted 13 months of fieldwork at these three sites, collecting field notes totaling more than 200,000 words. We conducted in-depth interviews with 22 tourists who visited wanghong cities in the first period, and 26 short-video producers or live streamers (18 of whom had physical travel experience in Xi'an) in the second period, from December 2020 to December 2021, during the COVID-19 pandemic. The main topics covered in the interviews included the following: tourists' perceptions, impressions, and viewpoints of wanghong cities; their personal experiences, cognitive feelings, and media practices during the visit; and live streamers' expectations for wanghong cities and how these expectations were met during COVID-19. All interviewees remained anonymous. Each interview lasted about 60 min and was subsequently fully transcribed verbatim. Both authors analyzed all transcripts with the assistance of NVivo. The NVivo software helped us to identify the main keywords mentioned by the interviewees, including short videos, clocking in, wanghong spots, wanghong foods, wanghong characters, live streaming, online tourism, Tang Dynasty culture, etc. Accordingly, the authors coded and analyzed the interview content relevant to the research questions. We also analyzed the texts the interviewees posted on social media to investigate how tourists carry out sightseeing in wanghong cities with certain media practices (see Table 1).

**Table 1.** Information form for interviewees in this study.

| Period 1 (December 2017 to May 2019) | | Period 2 (December 2020 to December 2021) | |
|---|---|---|---|
| Interviewee Code | Personal Information (Gender/Age/Job) | Interviewee Code | Personal Information (Gender/Age/Job) |
| A | F/20/College student | 1 | F/37/Teacher |
| B | F/25/Clerk | 2 | M/28/— |
| C | F/54/Nurse | 3 | M/28/Salesman |
| D | M/35/Taxi driver | 4 | F/37/— |
| E | M/25/College student | 5 | F/23/College student |

**Table 1.** *Cont.*

| Period 1 (December 2017 to May 2019) | | Period 2 (December 2020 to December 2021) | |
|---|---|---|---|
| Interviewee Code | Personal Information (Gender/Age/Job) | Interviewee Code | Personal Information (Gender/Age/Job) |
| F | F/25/— | 6 | M/33/Taxi driver |
| G | F/27/Master's student | 7 | M/—/Taxi driver |
| H | M/26/Master's student | 8 | M/29/Clerk |
| I | M/55/Professor | 9 | M/26/Courier |
| J | F/38/Teacher | 10 | F/37/Courier |
| K | M/37/Courier | 11 | F/19/Student |
| L | M/37/— | 12 | F/24/College student |
| M | F/25/— | 13 | F/31/— |
| N | M/32/Teacher | 14 | F/35/— |
| O | F/35/Teacher–student | 15 | F/25/Master's student |
| P | M/45/Restaurant owner | 16 | F/24/Master's student |
| Q | M/24/Restaurant waiter | 17 | M/24/Master's student |
| R | M/26/Clerk | 18 | M/30/Teacher |
| S | M/24/Student | 19 | M/34/Teacher |
| T | F/14/Student | 20 | F/35/Teacher |
| U | F/17/Student | 21 | F/—/Clerk |
| V | F/22/College student | 22 | M/16/Student |
| | | 23 | M/17/Student |
| | | 24 | M/—/College student |
| | | 25 | F/30/Clerk |
| | | 26 | F/38/Teacher |

## 4. Results

### 4.1. Clocking in and the Transformation of Xi'an

The "clocking in" phenomenon, whereby tourists upload images of their trips to social media platforms such as TikTok, is a key reason for the emergence of wanghong cities [1]. For tourists, clocking in means marking their physical presence and sending their location to social media platforms by taking pictures on their cellphones [5,6,28,79]. This is a highly participatory type of city tourism, in which tourists not only perceive the city with their bodies but can also strengthen their connection with the city through personalization. In the interviews, one tourist said:

> Traveling used to be all about seeing things and taking pictures. When I start clocking in and pinpoint my location on my phone, I feel like I know better where I am, and because the name of the place is familiar, I feel like that place is connected to me. (Interviewee L, December 2018)

Therefore, one of the meanings of clocking in is identifying a personal connection with the destination through electronic map positioning; combined with their posted text, this tourist edited the location information as "Xi'an, at the foot of the 600-year-old City Wall." This means that clocking in requires tourists to demonstrate a personal understanding of the destination rather than just taking pictures for the sake of remembering. Therefore, tourists usually upload their clocking in information to social media platforms with their interpretation of the destination for others to read. Indeed, many previous studies have ex-

plored the role and influence of the Internet, especially social media, on tourists' destination choices and decision-making [5,6,79]. In this sense, clocking in works, as user-generated content on social media helps cities become popular destinations for sightseers [7].

> When I post, I always choose something meaningful to post and then add a sentence or two to it. Like when I went to Datang Everbright City, I posted a video with "If you have not seen the Great Tang dynasty, come to Datang Everbright City, you will see!" (Interviewee N, December 2018)

Before social media existed, people took photos during their travels to remember their experiences or share them with friends and relatives, which shows the intuitive connection between photography and tourism [80]. With the popularity of social media, tourists' photos have gained value and importance and have even influenced perceptions of destinations [81]. Recently, clocking-in culture has turned the relationship between tourists and photography upside down, so that tourists may travel to a destination just to take a photo at a wanghong spot. Therefore, many cities have undergone a visual transformation to make them more attractive to tourists, and especially to the clocking-in culture. Previous studies indicated that the influence of online information and offline tourism is cyclical [82], but the actual effect of online information on tourist destinations is still relatively under-researched. Therefore, the following section focuses on how Xi'an transformed into a wanghong city that caters to clocking-in culture to explore this influence and effect further. As a historical capital with modern sights, sounds, and electronic decoration, Xi'an has become a visually Tang-themed city. More specifically, we can see at least three visual reconstruction and transformation aspects: landscapes, people, and performances.

To enable tourists to take exquisite photos and videos, many streets in Xi'an have been decorated and illuminated with red lanterns. Near landmarks such as Datang Everbright City and Datang Furong Garden, in particular, lanterns can be seen everywhere, giving tourists the feeling of going back to the Tang Dynasty. Due to the good photographic conditions, these places are visited by the most tourists when they travel to Xi'an.

> As long as I clock in here and do not display the position data, others will know I am in Xi'an. Because the scenery and decoration here look very special, just like the feeling of Chang'an in the Tang Dynasty. (Interviewee V, May 2019)

With the creation of such a striking scene, people can reinterpret the history and presence of the city and reconstruct the cityscape of Xi'an in the Great Tang Dynasty [2]. To enhance the tourist attraction of Xi'an as a wanghong city, not only has Tang scenery been built in places frequently visited by tourists, but the transformation of the whole city has created a Tang-style theme park (see Figure 1). This is reflected in some common street buildings, lights, and decorations, which seem to transport Xi'an into the Tang Dynasty. Whether visiting the place in person or on the Internet, tourists are entranced by the phantasmagoria of the Tang style.

Besides the transformation of the landscape, the people in this city environment have also been unknowingly involved in Xi'an's transformation into a wanghong city that caters to tourists' clocking-in culture. Many people in Xi'an, especially in wanghong spots and streets, walk around in Hanfu (汉服) during the festival season. In Hanfu, people generally greet each other with traditional etiquette when they meet.

> Because here, many people will dress like this. I saw many people on the internet wearing Hanfu here. I was attracted by this atmosphere, so I chose Xi'an as the destination for this trip. I just wanted to see if it was like the video. Because Xi'an is an ancient capital, you can see that these streets look very classical, so there is nothing special about wearing these ancient costumes here, and the photos were taken better, the more likes I get. (Interviewee U, May 2019)

In this sense, the driving force behind wanghong cities is mainly their visual attraction on the screen; the screen effect is the key to attracting tourists. Large LED screens and mobile smart screens for tourists abound in modern cities, and can be found in every corner,

promoting their visual transformation [59]. Tourists have become accustomed to seeing cities through their screens. Therefore, the entire city environment has been redesigned to fit the screen and satisfy tourists' need for visual images and videos.

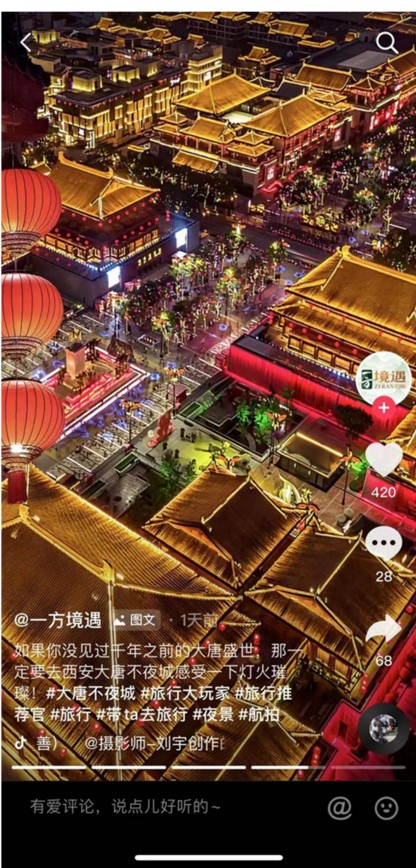

**Figure 1.** Datang Everbright City. Text: "If you have not seen the Great Tang Dynasty a thousand years ago, go to the Datang Everbright City in Xi'an to experience the bright lights!" Source: TikTok images captured by the authors with screenshots. The TikTok account here is @Situation of one side.

In terms of performance, the most typical case is the tumbler performance (不倒翁表演) in Datang Everbright City. The performance in 2019 inspired many imitators and had more than 230 million views on TikTok, and many people traveled to Xi'an just to see the actress perform. After seeing this actress's contribution to urban tourism, Xi'an and other wanghong cities have begun to consider how to construct wanghong spots to attract tourists. As another example, after the outbreak of COVID-19, many scenic spots were closed and people were restricted from traveling. Therefore, during the Spring Festival in 2021, there was a special performance of a 3D light show at the annual Lantern Festival at Xi'an City Wall, with the theme of "The Most Chinese, See Xi'an" (最中国,看西安). Dressed in Tang Dynasty armor, warriors sailed across the City Wall in spaceships, creating a time-traveling viewing experience. The show was mainly designed for video transmission, so that, even though there were very few offline tourists at the time, the light show captured mass attention online through the dissemination of short videos.

The visual culture of short videos has been crucial for Xi'an's transformation into a wanghong city. During the COVID-19 pandemic, many Chinese cities began to market their urban image through short videos. For example, Xi'an created a series of related hashtags on TikTok such as "#Hi Go Xi'an," "#What to eat in Xi'an," and "#Xi'an tour following TikTok", which received widespread attention and engaged active participation. These hashtags were mainly used to guide people to share their personal views of the city and creative productions on the social media platform so that the city could be seen and remain popular even during the pandemic lockdown. It is worth noting that the hashtag "#Xi'an

tour following TikTok" has been viewed 1.85 billion times. The introductory words given for this topic are as follows (see Figure 2):

> What comes to mind about Xi'an? Is it the majesty of the Terracotta Army, the morning bell and evening drum of the Bell and Drum Tower, the solemnity of the huge Wild Goose Pagoda, the cupping wine in Yongxing Square, or the local national dish Rou Jia Mo, Mutton Bubble Mo . . . ? Have you been to Xi'an before? What do you miss about food, scenery, or culture? If you are in Xi'an or currently traveling to the city, you may want to take in the city's special features with TikTok and let us experience the new charm of this ancient cultural capital together.

This introduction mentions the famous sights of Xi'an and asks tourists to capture the new charm of the ancient capital in short videos. The purpose of the hashtags is to encourage tourists to clock in and keep the city popular; a study suggests that newly developed media marketing methods have created a spillover effect between online and offline travel [83]. With this hashtag, user-generated content is not limited to the food, scenery, and landscapes mentioned in the text, but also personal summaries of tourism strategies, imaginative personal stories of the city, and personal participation in the record of wanghong events.

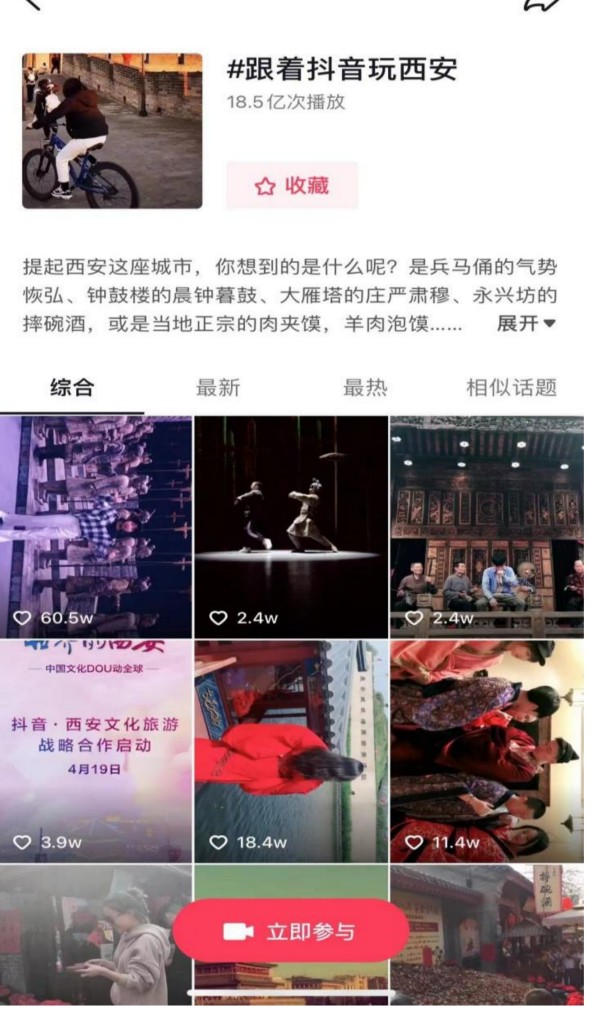

**Figure 2.** Hashtag "#Xi'an tour following TikTok". Text: "When you think of the city of Xi'an, what do you think of?" Source: TikTok images captured by the authors with screenshots.

*4.2. Live Streaming and Consuming Xi'an*

The phenomenon of wanghong cities has led to changes not only in terms of the urban transformation described above, but also in people's travel decisions and consumption. It has been argued that the wanghong economy, as a form of daily entertainment, has a great impact on the revitalization of the local economy [84], which can convert audience traffic into money through online commerce and advertising [85,86]. Wanghong cities also revitalize the economy, while being characterized by live streams, an important form of the wanghong economy in China.

Wanghong cities promote the live streaming of tourism, especially food exploration and explanations of tourist attractions and local life, which in turn attracts tourists' interest. During the COVID-19 pandemic, tourists and organizers preferred low-risk travel. One such low-risk method involves tourists traveling with smartphones and watching live streams, which extends the tourism industry chain and creates a new marketing model [27]. Live streaming works together with short videos to satisfy tourists' need to experience cities without actually being there. Thus, with the help of live streaming, Chinese tourism developed a new consumption model to reduce economic losses during the COVID-19 pandemic. In the last research period, we found that live streaming influences the urban tourism economy in three main ways: tourists are induced to travel to the city, to buy local goods based on the live stream, and to travel to the city with the live stream:

> I like to watch the live stream of Datang Everbright City because there are so many retro shows. When I saw Feng Jiachen's performance of the tumbler, I decided to go to that city in person. (Interviewee 2, December 2020)

Among the respondents, four of them (respondents 8, 15, 16, and 20) clearly expressed that they decided to travel to this city because they saw a live stream. These respondents mentioned that these live streams further fueled their desire to visit wanghong cities. Previous studies have shown that social media is effective in providing comprehensive destination information and updating online information [79,82]; similarly, live streaming in wanghong cities has convinced some tourists to visit. Furthermore, live streaming can directly persuade tourists to return to tourist spots through the second type of consumption, buying local goods through the live stream (see Figure 3):

> I started live streaming soon after I came to Xi'an to study because Xi'an is a famous city on the Internet, so sometimes many people watch my live stream. I mainly explore the stores and visit some restaurants to live stream my taste test. Sometimes I talk about how this food tastes, where it comes from, and so on. Since I have more followers, I have been working with companies to promote their restaurants or their products. The best thing I ever did was selling chili oil sauce for a pasta store. I sold over 300 bottles in one hour. (Interviewee 12, September 2021)

Compared to traditional tourism consumption, live stream consumption revitalizes the city's tourism economy through creative media practices, especially in the context of the COVID-19 pandemic:

> I like to buy something local while watching live streams. It makes me feel like I am following someone else's journey, but I can also try local food or buy local souvenirs. This way, I feel like I can somewhat satisfy my regret of not being able to travel offline. (Interviewee 5, April 2021)

In this way, we can understand that viewers discover not only the local landscape from a new perspective through online travel but also the local characteristics by mail, so that they can experience the indispensable connections of physical tourism, such as tasting local food and buying souvenirs. When viewers receive a chili oil with local flavor or a cute Tangie doll with local characteristics, the specificity of the destination can directly touch their real life in material form and arouse their desire to travel in person. In this way, the interaction between viewers, tourists, media, and the city can be reflected in a direct

material form in wanghong cities in real life, rather than virtually on the Internet or in distant locations.

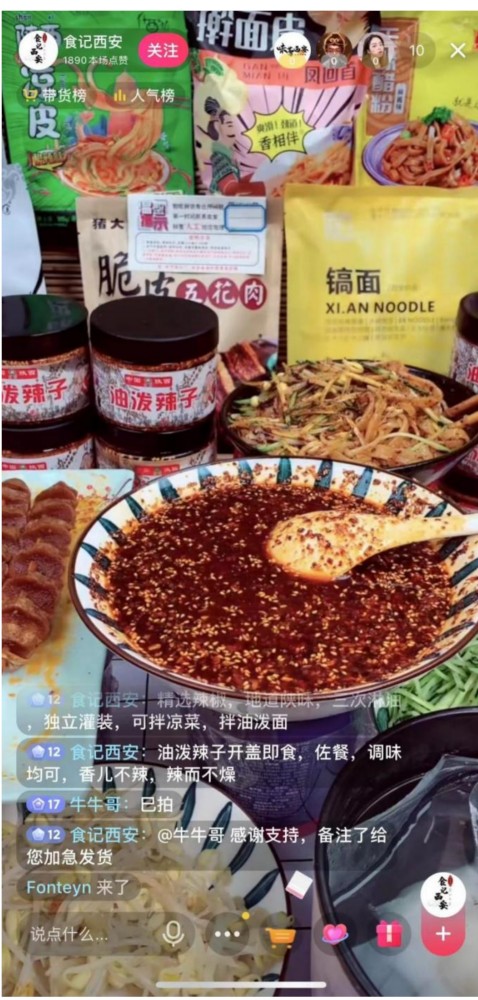

**Figure 3.** Live stream of Xi'an local cuisine. The text shows interactions between viewers and the live streamer. Source: TikTok live streams captured by the authors with screenshots. The live streamer in this screenshot is @Recording foods in Xi'an.

Although the Internet has been considered a prime information source by tourists over the last two decades [85,87], it is usually used for reference rather than making decisions. Live streaming, however, is more persuasive, and tourists are more willing to travel based on live streaming because live streamers are usually trusted to attract potential tourists. For example, there are popular live streaming themes with ingenious perspectives, such as "Take you to experience a different ancient capital, Xi'an" and "Only local people in Xi'an know these shops in the alleys, so let us try them today". In our fieldwork and interviews, we found that live streaming even acted as a navigation aid for some tourists:

> I watched a lot of live streaming about Xi'an before this trip and felt like I was very familiar with this city. However, when I am here, I still want to turn on the live streaming and see what's here. That way I can get to know the city better. I just went to a restaurant that is known to the locals and that I would not know if I were looking for it myself. (Interviewee 8, May 2021)

### 4.3. Cyberflaneurs in Xi'an during the COVID-19 Pandemic

During the COVID-19 pandemic, when the idea of "little contact, less gathering" gradually became the new normal, traveling and gathering among the Chinese population

greatly decreased. Even though the pandemic has been effectively controlled, consumers still rationally weigh the risks of travel and avoid large gatherings, and thus the risk of infection. Accordingly, the tourism industry also changed significantly during the pandemic. The number of physical tourists declined sharply, and, accordingly, tourists have discovered the benefits of Internet tourism. Tourists not only travel online through short videos and live streams but also experience their favorite destinations in more detail by shopping online for local items through live streaming. Therefore, Chinese tourists did not stop traveling during the COVID-19 pandemic, but rather adopted cybertravel in the form of cyberflaneurs in the new media environment.

On this basis, Benjamin proposed the concept of the flaneur: someone who wanders the city to realize its importance [88]. The flâneur became a symbol of modernity in 19th-century Paris. In Gibson's science fiction novel *Neuromancer*, people are removed from their physical existence to interact with the Internet, imagining the wanderings of future Cybermen. This kind of wandering reflects how, in a networked and mediated society, the virtual experience comes first and the body comes second. Accordingly, some scholars are attempting to develop a new concept of the cyberflaneur [87,89] to describe the relationship between people and media cities and to link cyber cities with physical cities.

The emergence of wanghong cities against the backdrop of the COVID-19 pandemic has highlighted the increasing importance of the cyberflaneur, as people tend to turn to cybertourism to minimize physical travel and avoid the risk of contracting the virus [90]. The cyberspace flâneur maintains a looser relationship with the physical environment [77], which makes cybertravel a better option than on-site travel:

> I love to travel, but I have not traveled much since the COVID-19 outbreak. Because of the pandemic, it is very difficult to travel, and now it is convenient to travel online. I often watch short videos and live streamings about the tour. Surprisingly, I have even discovered things that I could not see when I was physically traveling before. (Interviewee 20, November 2021)

This is similar to Urry's [91] statement: "People are tourists most of the time, whether they are literally mobile or only experience simulated mobility through the incredible fluidity of multiple signs and electronic images." Indeed, the development of media technologies has enriched online tourism [92]; as more people get involved in online tourism, more user-generated content fills social media platforms, turning them into cyberspaces that are relevant to but different from real travel destinations, through which cyberflaneurs can see more diverse tourist destinations.

There has been a particular increase in content on short-video platforms with hashtags such as #Yunyou Shaanxi and #Chaowan Xi'an ("Make fun of stylish Xi'an"). There are online games for virtual makeovers, inviting cyberflaneurs to imitate and tag for personal interest (see Figure 4). These two hashtag activities popularized cyberflaneurism in Xi'an during the COVID-19 pandemic. Some reminder texts were also created on these platforms to prompt interactions between cyberflaneurs. For example, one participant posted a video of herself traveling before the pandemic, writing, "I traveled to Xi'an in the summer of 2019, before the pandemic. . . . Looking forward to checking back in when the pandemic is over!" This struck a chord with many people. One said, "I was looking forward to going to Xi'an, but I do not have time to go there. I will go there when the pandemic is over." Such interactions are also an important part of cybertours. Although cyberflaneurs cannot directly reinforce preferences for destinations by watching short videos or live streaming in virtual reality, which VR technology would allow [41], sharing and discussing travel experiences is also a way to reinforce the impact of destinations.

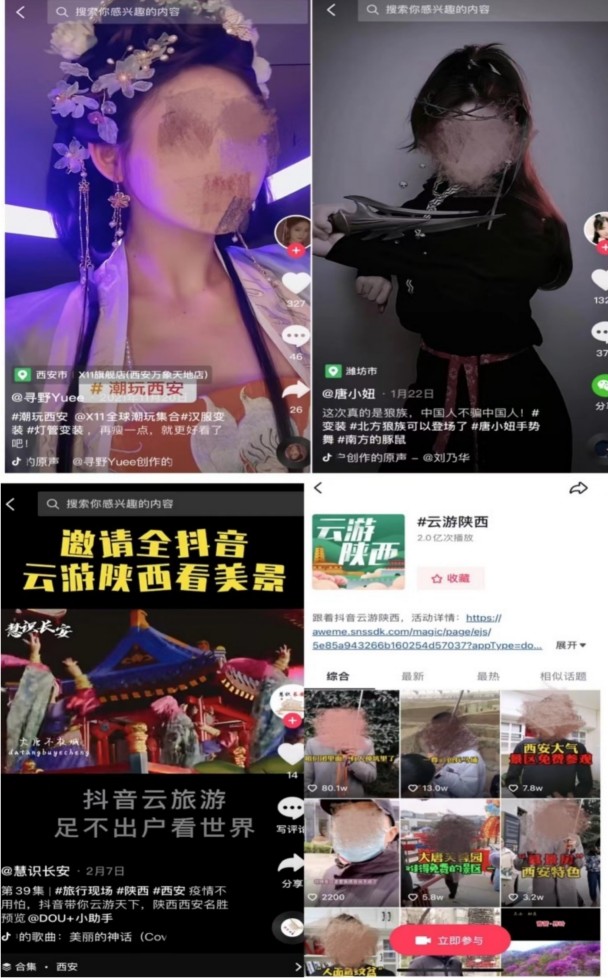

**Figure 4.** The new period of cybertourism. Text: introductions of online games for virtual makeovers and the #Yunyou Shaanxi hashtag. Source: TikTok images captured by the authors with screenshots. The accounts and hashtag in these screenshots are @Search Yuee, @Miss Tang, @Knowing Xi'an Smartly, and #Touring Shaanxi online.

## 5. Discussion

Tourism is considered an important part of modern human life. Richards points out that contemporary tourism has become a way of life in everyday culture, not just in relation to sightseeing [93]. This has changed in the era of new media, and has been challenged by the COVID-19 pandemic. The results shown in this paper suggest that there is a trend toward the mediatization of urban tourism against this new backdrop, including the mediatization of cities to become wanghong cities, the mediatization of interactions between tourists and cities, and the mediatization of tourists themselves.

First, the city of Xi'an has undergone a visual transformation to make itself more attractive to tourists within the clocking-in culture. City streets, characters, and performances have been visually transformed to make it easier for visitors to take clocking in videos, and the previously modern city is filled with phantasmagoria evoking the ancient Tang Dynasty. When clocking in becomes an important channel in the process of urban mediatization, media is no longer just a form of technology that people choose to use when traveling or promoting a city, but a structural condition [94]. By "clocking in" the wanghong attractions and foods in a city, tourists interact with the city and leave their mark on it. These media practices suggest that the media content on urban tourism produced by tourists embodies individual cognition and understanding of tourist destinations, which opens up a creative space that allows tourists to define a new image and change the original meaning of cities.

Second, there are also changes in the consumption patterns of tourists when cities are transformed into wanghong cities. Through the media practice of live streaming, tourists not only experience cities remotely and buy local commodities, but also view the live streaming content as popular travel guidance. For tourists of wanghong cities, traveling with live streaming is a special experience, and the live streaming directly affects which sights or restaurants they want to visit. During the field research, we found that live-streaming wanghong restaurants are more popular than those with established brands, especially among young people, who prefer restaurants recommended in live streams. Media content, especially in live streams, is used as reference information for people's travel planning. More importantly, it plays a leading role in tourism, which to some extent determines tourists' destination choices and consumption.

Third, the emergence of wanghong cities against the backdrop of the COVID-19 pandemic has activated a new digital tourism pattern of cyberflaneurs.

This suggests that tourism has less to do with mass versions of cultural experiences and more to do with photographs, videos, and subsequent performances and interactions on social media platforms. These new media practices are creating a new Xi'an in cyberspace. This is in response to the regret of not being able to visit the place in person, so that people can find interest in being cyberflaneurs of this city. However, if people rely too much on the online experience, offline urban tourism will become redundant, and the lack of physical contact between tourists and cities will cause new problems. Therefore, the question of how to extend the advantages of wanghong cities to offline tourism will be an important research focus in the post-COVID-19 era.

Many offline tourist attractions in China have been temporarily closed due to the pandemic, and even after reopening, strict COVID-19 containment requirements have made offline travel troublesome. Therefore, online travel has become an important way for tourists to travel in the post-COVID-19 era, and cyberflaneurs take pleasure in interacting with wanghong cities through short videos or live streams. To explain such changes and challenges, this study proposed an innovative framework of tourists–media–cities (ToMeCi), which has its roots and paths in tourism, media, and urban studies, but goes beyond the traditional concepts of media and city, tourism and media, and tourism and city, to highlight the mediatization of cities and tourism against the backdrop of the COVID-19 pandemic. This framework engages the complex relationship between changes in media, city, and tourism on the one hand, and changes in various aspects of culture and society on the other. This framework emphasizes the role of the media, regarding it as part of a dynamic process that shapes social interactions and forms of communication, influences the logic of transformation of tourism and cities, and ultimately triggers changes in social structures. The comparison between the interviews conducted in the first and second periods shows that, after the pandemic, tourists are more eager to determine their travel goals by online recommendations and have a higher acceptance of online tourism, and more people have begun to interact with distant cities or people through media. In this sense, this framework is intended to be media-centered to engage constructively with researchers in tourism studies and urban studies who come to the table with tourism-centered or city-centered concepts and frameworks.

## 6. Conclusions

This study focuses on the phenomenon of wanghong cities against the backdrop of the COVID-19 pandemic, using Xi'an as a case study; our research showed how the city interacts with tourists through media transformation, how tourists' media practices have changed Xi'an and the tourism industry, and how the phenomenon of wanghong cities offers new opportunities for the entire tourism industry in the pandemic and in the post-COVID-19 era.

This triadic ToMeCi framework illustrates the dynamic mediatization relationship between media, tourists, and cities that underpins the emergence and development of wanghong cities. However, we must admit that the ToMeCi concept is not entirely new,

especially since the factors of "Me" (media) and "Ci" (city) have been sufficiently discussed by other scholars. However, the COVID-19 pandemic and creative media practices make us aware of the increasing importance of the "To" (tourist) factor. Tourists are no longer just viewers of city images through the media; they use their own media practices to significantly influence the tourism industry and city images. Specifically, tourists vote directly for their favorite destinations by clocking in, live streaming, and taking cybertours. This influences the decisions of other tourists and even the urban transformation of wanghong cities.

Based on our long-term field work in Xi'an, we believe that the new ToMeCi tourism model provides an appropriate theoretical summary of the marketing achievements of this wanghong city. It also inspires us to better develop urban tourism in the post-COVID-19 era and the social media age. First, post-COVID-19 urban tourism must prioritize the relationship between tourists and the media; in particular, it must encourage and stimulate tourists to participate in the construction of a city's image through new media practices. Second, more creative marketing strategies for urban tourism based on social media need to be developed, since social media has become the infrastructure of wanghong cities in the process of urban mediatization. Finally, it is worth considering the development of interactive and participatory projects through which tourists can participate in digital urban tourism with diverse new media.

Although the ToMeCi framework is based on our long-term field work in Chinese wanghong cities, it still needs further empirical research, especially that which uses quantitative or mixed research methods. Moreover, online tourism has developed in many countries around the world in the post-COVID-19 period. Therefore, case studies from other countries could provide new and different insights that would improve this theoretical framework.

**Author Contributions:** Conceptualization, T.F. and Z.L.; literature review, T.F.; methodology, T.F. and Z.L.; development of research results, T.F. and Z.L.; investigation, T.F. and Z.L.; preparation and review of the article, Z.L. and T.F. All authors have read and agreed to the published version of the manuscript.

**Funding:** The authors received no financial support for this research.

**Informed Consent Statement:** Informed consent was obtained from all subjects involved in the study.

**Data Availability Statement:** The data presented in this study are available on request from the corresponding author. The data are not publicly available due to confidentiality assurance of each participant's information.

**Conflicts of Interest:** The authors declare no conflict of interest.

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
