# Peer review of "Toward Tourists–Media–Cities Tourism: Xi’an as a Wanghong City"

_sustainability, doi:10.3390/su141911806_

Round 1

Reviewer 1 Report

The topic selection of this paper is novel and  interesting, but it lacks of theoretical research contributions, which leads to a flat and straight description of the whole paper. The knowledge introduces a phenomenon, so the academic value is not high.

In the abstract, the main conclusions of this paper need Valid statement rather than described. Where is the innovation?

In the introduction part, add the theoretical analysis of the influence of network development on the cognition of urban tourism image, now this paper only describes this phenomenon.

In the literature review section, what is the theoretical framework of this paper?

The picture information in this article is in Chinese, and most readers may not know it. How to deal with it needs to be considered.

This phenomenon is not considered in the theoretical development and macro framework of urban tourism image

The conclusion part lacks effective in-depth content and inspiration. The research is insufficient and vague.

Author Response

We appreciate the opportunity to revise our manuscript entitled “Toward Tourists-Media-Cities Tourism: Xi’an as a Wanghong City,” and we thank you for taking the time to provide such insightful guidance.

We carefully considered your comments as well as those offered by the three reviewers. Below, you will find revisions and replies to the comments made in response to the manuscript. We assigned numbers to the comments (in italics) so that we could appropriately address and cross-reference each specific issue/comment/question. Responses are below each comment in regular (non-bold) font. Within the responses, italicized sections denote changes we made to the manuscript itself. Within the manuscript itself, changes are highlighted in yellow. Where we made pervasive or exceptionally lengthy changes, we ask that you please refer to the manuscript itself. We thank you for your efforts in enhancing the quality of the manuscript and hope that the below revisions/responses satisfactorily address your concerns.

Thank you again for your effort and time. We look forward to hearing from you soon regarding this revised paper.

Sincerely,

The Authors

**************************************************************************************************

Editor’s comments: Your manuscript has now been reviewed by experts in the field.

Please revise the manuscript according to the referees' comments and upload the revised file within 10 days.

Response: We are pleased that the editor offered us an invitation to revise our work. Additionally, we appreciate you assigning three qualified reviewers to our manuscript. Their efforts and insights were a tremendous help to us during this revision. We sincerely appreciate the opportunity to revise our work for consideration for publication in Sustainability. We hope our revision meets your expectation. We next detail our responses to each reviewer’s concerns and comments.

Reviewer #1:

The topic selection of this paper is novel and interesting, but it lacks of theoretical research contributions, which leads to a flat and straight description of the whole paper. The knowledge introduces a phenomenon, so the academic value is not high.

Response: Thank you for taking the time to offer critical comments. In the following sections, you will find our responses to each of your points and suggestions. We are grateful for the time and energy you expended on our behalf.

COMMENT 1:

In the abstract, the main conclusions of this paper need Valid statement rather than described. Where is the innovation?

â–ºRESPONSE:

Thank you very much for providing us with constructive comments. We have rewritten the abstract to highlight the theoretical innovation of this paper, and now the content of the abstract is:

Abstract: This study investigates the phenomenon of Wanghong Cities in China to illustrate the dynamic relationships between media, tourists, and cities in the new normal of the post-COVID-19 era. Specifically, this study proposes the innovative analytical framework of “ToMeCi” (Tourists-Media-Cities), which is grounded in tourism studies, media studies, and urban studies while going beyond traditional concepts and previous studies of media city, tourism & media, tourism & city. Based on the case study of Xi’an, one of the most famous Wanghong Cities in China, this study attempts to answer the following research questions that how the deployment of creative media practices can create new digital tourism patterns in a specific Chinese context of Wanghong Cities, why it is reasonable and possible, and what is its implication. For the purpose of this study, a qualitative research method was employed that we conducted online and offline ethnographic fieldwork, textual analysis, and in-depth interviews with 22 tourists and 26 short video producers or live streamers. The findings reveal that the city of Xi’an transform itself into a Wanghong City to attract tourists who interact with the city through specific media practices of Clocking In and live streaming, with new the digital tourism pattern of Cyberflaneur emerged against the specific backdrop of the COVID-19 pandemic. The study finally discusses its possible contributions and limitations.

COMMENT 2:

In the introduction part, add the theoretical analysis of the influence of network development on the cognition of urban tourism image, now this paper only describes this phenomenon.

â–ºRESPONSE:

Thank you for your constructive comments. We have added the theoretical analysis of the influence of network development on the cognition of urban tourism image to support the introduction part. Please kindly refer to pages 1 to 2. They now read:

Underpinned by diverse creative media practices, we have witnessed the emer-gence of Wanghong Cities in China since 2018 [1]. Many cities like Xi’an, Chengdu, Chongqing, and Changsha have been labeled as Wanghong Cities because they share the criteria of Wanghong Cities which become famous online thus tourists tend to choose these cities as their travel destinations offline [2]. During this process, creative media practices like Clocking In exert a great impact on the phenomenon of Wanghong Cities. We have already become very familiar that tourists share their personal travel experiences on Weibo, Honeycomb, Fliggy, Facebook, and other social media platforms, and interact with followers and people with the same travel interests, through which tourists can modify the style of a city image through personal media practices, creating and participating in urban tourism marketing activities [3]. Indeed, there are many studies on the relationship between media practices and urban tourism [4-7], especially on how popular social media platforms have exerted great impacts on the cognition of urban tourism image [8-10], manifesting in the way tourists create an image of the destination [11]. As Govers, Go and Kumar argue, “technological advancement, the global media, and increased international competition affect the way in which desti-nations are imagined, perceived, and consumed. Image formation is no longer a one-way ‘push’ process of mass communication, but a dynamic one of selecting, re-flecting, sharing, and experiencing” [11] (p. 978). Recently, new media technologies such as social media, information and communication technologies (ICT), augmented reality (AR), and virtual reality (VR) have become increasingly important for the tour-ism industry, not only for destination marketing but also to help tourists choose a des-tination and avoid tourism risks [12-14].

Especially in the new backdrop of the COVID-19 pandemic which affects every part of the world, tourism an environmentally sensitive industry across the world has been hit hard by the coronavirus outbreak [15,16]. Researchers generally believe that the COVID-19 pandemic hurts tourism because of the lockdown, isolation, quarantine, and travel restrictions caused by the pandemic [17,18]. Furthermore, non-pharmacological measures like limiting the flow of tourists, taking temperatures, wearing masks, and showing health codes in scenic spots also affect tourist activities especially those of offline travels [19,20]. Facing such challenges, new media practices based on new media technologies such as AR, VR, live streaming, and short video have been argued to create new opportunities for the tourism industry and promote new tourism patterns like virtual tourism [21-23]. According to the data center of the Min-istry of Culture and Tourism, during China’s National Day holiday in 2021, from Oc-tober 1 to 7, the number of domestic tourism trips decreased by 1.5% year on year, and the domestic tourism income decreased by 4.7% year on year, but many scenic spots in Wanghong Cities such as Datang Everbright City in Xi’an attracted more people than in previous years [24]. Therefore, the study of Wanghong Cities and the theoretical framework and practical mechanisms behind them may have some reference value for thinking about the reopening of scenic spots and the restoration of tourism as a whole in the period after COVID-19 [18].

COMMENT 3:

In the literature review section, what is the theoretical framework of this paper?

â–ºRESPONSE:

Thanks for your comments. We have substantially revised the literature review according to your suggestion to more explicitly highlight the theoretical framework of ToMeCi we proposed. Please kindly refer to pages 2 to 5. The main part of the revision now reads:

Focused on the new Wanghong City phenomenon, this study would like to develop a triadic theoretical framework of ToMeCi (Tourists-Media-Cities) to rejuvenate the relationship between tourists, media, and cities. Indeed, in recent years, the important phenomenon of Wanghong Cities has attracted the interest of scholars in the fields of tourism studies, urban studies, and media studies [27-29]. But this study would like to go beyond the traditional studies on the relationships between tourists and cities, media and cities, tourists and media, to further explore the dynamic inter-activity of tourists-media-city with the innovative framework of ToMeCi, which has its roots and routes in urban tourism studies, as well as media and tourism studies. We would tease up these previous studies below.

There have been studies on urban tourism that focus on the relationship between tourism and the city, focusing on the impact of tourism on cities [30,31]. On the one hand, tourism can promote urban economic development and infrastructure construction, and improve the urban employment rate, which are positive effects of urban tourism [32]. On the other hand, many tourists coming to urban tourism have brought traffic congestion, pollution, and other problems [29]. Studies on urban tourism also focus on tourists and locals, examining locals' views on tourism development in their cities and how visitors move around the city [33-35]. Overall, creative and historical cities such as Xi'an have a higher probability of becoming tourist destinations [36-38]. This kind of city tourism is growing rapidly in the tourism market because of the rapid urbanization, low travel costs, the increase in short vacations, and popular online booking [39].

In addition to the relationship between tourism and the city or media and tourism, the Wanghong Cities phenomenon also leads to the relationship between media and the city, which is relatively scarcely studied in the field of tourism studies and even urban tourism studies. However, the relationship between media and the city has long been a critical topic in media studies with a specific research strand of Media City [26,50]. This strand of research assumes that the city is originally a medium [51]and that the contemporary city is a media-architecture complex [26]. From this perspective, scholars focus on research like urban digital infrastructure [52], mobile media lives in urban tourism [53], media’s impact on citizens [50,54,55], and geomedia in cities [50]. More recently, the Media City studies have continued to develop with contributions from more diverse disciplines and can be roughly divided into two approaches, namely, cities in media and media in cities.

However, these studies focused mainly on the binary relationship between media and city, as well as media and city tourism, ignoring the increasingly important role of tourists. Tourists, however, remain an important component of tourism research because they decide which attractions are worth seeing and sharing with other tourists. Especially in the era of social media, the image of a city and the tourism industry are no longer only constructed by the mass media but depend on each social media user. Especially in the context of Wanghong Cities, which rely heavily on the social media Clock In culture, through which tourists vote for the country with their media practices [2]. On the one hand, people are keen to travel to Wanghong Cities because they are influenced by photos or videos of others Clocking In on social media; on the other hand, Clocking In on the city’s popular Internet spots becomes an important part of a trip [1]. With tourists flocking to social media, one particular city stands out among many others. It has become a Wanghong destination for more and more people, where we can observe the increasing interactivity of tourists, media, and the city.

Therefore, focused on the new Wanghong City phenomenon, this study would like to go beyond the traditional studies on the relationships between tourists and cities, media and cities, tourists and media, to articulate them to develop a triadic theoretical framework of ToMeCi to rejuvenate the relationship between tourists, media, and cities. Indeed, In recent years, the important phenomenon of Wanghong Cities has attracted the interest of scholars in the fields of tourism studies, urban studies, and media studies [27,28,39]. Based on these previous studies, this study would like to further explore the dynamic interactivity of tourists-media-city with the innovative framework of ToMeCi, focusing on the following research questions: How does the city transform into a Wanghong city to attract tourists? How do tourists interact with the city through specific media practices (e.g., Clocking In and live streaming)? How do tourists virtually travel (e.g., live streaming) around Wanghong City during the COVID-19 pandemic? What new digital tourism patterns (e.g., cyberflaneur) emerge from these media practices, and how do they emerge against the specific backdrop of the COVID-19 pan-demic?

COMMENT 4:

The picture information in this article is in Chinese, and most readers may not know it. How to deal with it needs to be considered.

â–ºRESPONSE:

Your suggestions are very reasonable. We have given corresponding explanations on the information of these pictures, hoping to clearly explain the meaning of these pictures and the intention of using these pictures.

On page 8, Figure 1, We add text: If you haven’t seen the Great Tang Dynasty a thousand years ago, go to the Datang Everbright City in Xi’an to experience the bright lights!

On page 10, Figure 2, We add text: When you think of the city of Xi'an, what do you think of?

On page 12, Figure 3, We add text: Texts in live streaming are interactions between viewers and live streamers.

On page 14, Figure 4, We add text: Texts are the introduction of different online games of the virtual makeover and the #Yunyou Shaanxi hashtag.

COMMENT 5:

This phenomenon is not considered in the theoretical development and macro framework of urban tourism image

â–ºRESPONSE:

Thank you for your expectation of improving the academic value of our paper. We tried our best to situate our study in the theoretical development and macro framework of urban tourism image. Due to the space limit, we added a particular paragraph on page 2:

Indeed, there are many studies on the relationship between media practices and urban tourism [4-7], especially on how popular social media platforms have exerted great impacts on the cognition of urban tourism image [8-10], manifesting in the way tourists create an image of the destination [11]. As Govers, Go and Kumar argue, “technological advancement, the global media, and increased international competition affect the way in which destinations are imagined, perceived, and consumed. Image formation is no longer a one-way ‘push’ process of mass communication, but a dynamic one of selecting, reflecting, sharing, and experiencing” [11] (p. 978). Recently, new media technologies such as social media, information and communication technologies (ICT), augmented reality (AR), and virtual reality (VR) have become increasingly important for the tourism industry, not only for destination marketing but also to help tourists choose a destination and avoid tourism risks [12-14].

COMMENT 6:

The conclusion part lacks effective in-depth content and inspiration. The research is insufficient and vague.

â–ºRESPONSE:

We agree with the reviewer and have added a new section of the Discussion according to all three reviewers’ suggestions. Please kindly refer to page 15. It now reads:

  1. Discussion

Tourism is considered an important part of modern human life. Richards points out that contemporary tourism has become a way of life and everyday culture, not just sightseeing [93]. This kind of everyday life and culture has changed in the era of new media and has been challenged by the COVID-19 pandemic. The results shown above suggest that there has been a trend of the mediatization of urban tourism – including the mediatization of a city to become a Wanghong City, the mediatization of interaction between tourists and the city, and the mediatization of tourists themselves – against this new backdrop.

First, the city has undergone a visual transformation to make itself more attractive to tourists within the Clock In culture. When Clock In becomes an important link in the process of urban mediatization, media is no longer just a technology that people choose to use when traveling or promoting a city, but a structural condition of urban mediatization [94]. These media practices suggest that the media content about urban tourism produced by tourists embodies the individual cognition and understanding of the tourist destination, which opens up a creative space that allows tourists to define the new image of the city through particular media practices.

Second, there are also changes in the consumption patterns of tourists when the city is transformed into a Wanghong City. For tourists of Wanghong Cities, traveling with live streaming is a special travel experience, and the live streaming will directly affect which sights or restaurants tourists want to visit. During the field research, we found that live streaming Wanghong restaurants are more popular than those with established brands, especially among young people who prefer the restaurants recommended in the live stream. Media practices, especially live streams, are used as reference information in people's travel planning. More importantly, they play a leading role in tourism, which to some extent determines destination choice and consumption.

Third, the emergence of Wanghong Cities against the backdrop of the COVID-19 pandemic has activated a new digital tourism pattern of Cyberflaneur. It suggests that tourism has less to do with mass versions of cultural experiences and more to do with photographs, videos, and subsequent performances and interactions on social media platforms. These new media practices are creating a new Xi'an in cyberspace. It is a response to the regret of not being able to visit the place in person so that people can find interest in being the cyberflaneur of this city.

To explain such changes and challenges, this study proposes an innovative framework of "ToMeCi" (Tourists-Media-Cities), which has its roots and paths in tourism, media, and urban studies, but goes beyond the traditional concepts of media city, tourism & media, tourism & city. This framework engages with the complex relationship between changes in media, city, and tourism, on the one hand, and changes in various fields of culture and society on the other. But this framework emphasizes the role of the media, regarding media as a dynamic process that shapes social interaction and communicative figurations, influences the logic of transformation of tourism and city, and ultimately triggers the change of social structure. In this sense, this framework attempts to be “media-centered” to engage constructively with researchers from tourism studies and urban studies who come to the table with “tourism-centered” or “city-centered” concepts and frameworks.

Reviewer 2 Report

The article deals with a pertinent theme: the visibility of places in social media platforms and its relevance in a post-Covid tourism, and how this affects the triangular relationship between tourists, media, and the touristic city. There are, however, a set of adjustments that matters to carry out.

A more detailed description of the Wanghong Cities phenomenon would be useful, so the narrative line of the manuscript could be easier to follow: When it appeared? What are the criteria used to consider that a specific city is part of the Wanghong Cities?

Considering the centrality that the COVID-19 assumes in this paper, it would be convenient to have a clear picture of what have been the concrete consequences of the pandemic for the Chinese touristic cities, and how the non-pharmacological measures imposed by the Chinese Government affected touristic activity.

There are statements lacking evidence, such as the one which starts on the line 55: “… we noticed that during China’s National Day holiday in 2021, many scenic spots in Wanghong Cities especially Xi’an attracted more people than in previous years.”.

In the first paragraph of the second section (line 80) has sentences which are difficult to understand, and arguments are difficult to capture.

Regarding the Methods section, it would be important to have more information on the way the interviews were conducted: how were the respondents selected? What were the objectives of the interview? What were the main topics covered? Additionally, more information should be provided on the methods and techniques used to analyse these interviews.

I found difficult to understand what the main results of the research are. Also, it would be important to see the three questions presented on the abstract clearly addressed on the last sections of the manuscript. Moreover, I believe it would be important to have a “discussion” section where the results of the study should be critically interpreted. What are the contributions of the results to the current debates on digital spaces how this affects the interaction between tourists and the touristic city?

Abbreviations such as AR and VR should be defined at first mention.

The paper would improve considerably if the text would be divided in the following sections: Literature Review, Results, Discussion and Conclusions.

The References section needs to be arranged as per the guidelines of the journal.

Reviewer 3 Report

Overall, this is an interesting submission but not yet ready for publication. The following are the key problems.

1. The paper is over ambitious. It aims to answer three difficult and profound questions: “How does the city of Xi’an transform itself into a Wanghong City to attract tourists? How do tourists interact with the city through specific media practices (Clocking In and live streaming, for example)? What kind of new digital tourism patterns (Cyber-15 flaneur, for example) emerge during these media practices, and how do they emerge against the specific backdrop of the COVID-19 pandemic?

As a result, the focus of the paper is not prominent enough, the research is not deep enough, and the method is not scientific enough. Perhaps, it would be better to reduce ambition in order to better explore each of the objectives.

2. Literature review is insufficiently focused. Even though the authors refer 81 items, a big part of them are related to themes that are just slightly debated in the paper and seem to be on the list in order to make it look long. The 18 and 19 references, for example, are referred once in the paper as “studies on urban tourism that focus on the relationship between tourism and the city”. Both references focus on overtourism, a theme that is never treated in this paper. At the same time, many other important and influential papers and books that analyze the global relationship between tourism and the city are forgotten.

3 (and the most important problem) Conclusions are insipid, superficial and do not add anything substantially new to the knowledge of the relationship between Tourism, media and cities, which was, in fact, the main topic of this paper. In general, although this paper has certain research value, its theoretical significance is not clear enough, the content of the paper needs to be improved and the depth of the research needs to be further explored.

Round 2

Reviewer 1 Report

Please further enrich the innovative points, some of the language needs to be polished

Reviewer 2 Report

The manuscript has improved considerably, and the authors have clarified several of the questions I raised in my previous review. However, I consider the manuscript still requires major revisions, as both the logical coherence and the strength of the arguments must be enhanced. Please take into consideration the following issues:  

- Regarding the “Methods” section, although the criteria used to data collection are now clearer, the methodological process through which the analysis of the interview’s content went through is still unknown. Please clarify the approach used to analyze the texts;

- The “Discussion” section, as developed, is far from responding to the questions raised by the authors in the "Introduction" section: “How does the city transform into a Wanghong city to attract tourists? How do tourists interact with the city through specific media practices (e.g., Clocking In and live streaming)? How do tourists virtually travel (e.g., live streaming) around Wanghong City during the COVID-19 pandemic? What new digital tourism patterns (e.g., cyberflaneur) emerge from these media practices, and how do they emerge against the specific backdrop of the COVID-19 pandemic?” (lines 208–213). I strongly recommend the improvement of this section, to better its connection with the objectives presented both in the abstract and in the introduction. Furthermore, it would be interesting to see clearly explained what patterns and/or differences were identified among the results obtained in the two different research periods;

- The “Conclusion” section should be improved, and the authors need to elaborate what are the main findings of their study and their implications for the management of urban tourism destinations in a post-COVID-19 era. Moreover, considering the results presented, I found no evidence to support the following statement: “During the field research, we found that live streaming Wanghong restaurants are more popular than those with established brands, especially among young people who prefer the restaurants recommended in the live stream.”;

- Language improvements are needed. I strongly recommend the revision of the manuscript by an English speaker;

- The source of the figures presented must be mentioned.  

Round 3

Reviewer 2 Report

Dear authors,

After carefully reading the new version of your manuscript, I consider that it addresses the main questions I raised in my past reviews. However, I would like to ask your attention for the following issues:

-       The sentences between the line 567 and 571 are repeated in the line 576.

-      There are many long sentences in the text which are hard to read, for example: lines 200-203, 241-245, 386-389, among others. Please make them shorter and easier to understand.

-       Please check if the following expression corresponds to what you want to state “…through their mow media practice” (line 621);

-       Considering the recommendations introduced in the Conclusion section (lines 616-626), it would be interesting if you could provide your vision on the potential risks of making the urban tourism more dependent on the social media.
